# Magnetic Separation of Oxoacid of Boron from Salt-Lake Brine by Synergistically Enhanced Boron Adsorbents of Glucose-Functionalized SiO_2_ and Graphene

**DOI:** 10.3390/ijms231911356

**Published:** 2022-09-26

**Authors:** Qinglong Luo, Xueying Wang, Mingzhe Dong, Xueli Huang, Zhijian Wu, Jun Li

**Affiliations:** 1Key Laboratory of Comprehensive and Highly Efficient Utilization of Salt Lake Resources, Qinghai Institute of Salt Lakes, Chinese Academy of Sciences, Xining 810008, China; 2Key Laboratory of Salt Lake Resources Chemistry of Qinghai Province, Qinghai Institute of Salt Lakes, Chinese Academy of Sciences, Xining 810008, China; 3College of Chemical Engineering, Xinjiang University, Urumqi 830046, China

**Keywords:** adsorption, boron adsorbents, magnetic separation, SiO_2_, graphene, salt-lake brines

## Abstract

The adsorption separation and extraction of low-concentration boron from salt-lake brine have great significance. Magnetic separation avoids the problem of adsorbent granulation and improves the usage efficiency. The silicon-based adsorbents have attracted interest due to their superior acid and alkali resistance, in which polyhydroxy graphene enhances the adsorption of boron ions. Herein different boron adsorbents, derived by magnetic separation, were developed and characterized by SEM, TEM, XPS, VSM, FT-IR, and XRD analysis. The adsorption-desorption performance of boron adsorbents with different compositions was evaluated. The isotherms and kinetics parameters of the boron extraction were evaluated based on adsorption-desorption tests. The graphene-based magnetic adsorbent (Go-Fe_3_O_4_@SiO_2_@mSiO_2_-Glu) registered a high boron adsorption capacity of 23.90 mg/g at pH = 9 in the boron solution and 24.84 mg/g for East Taigener salt-lake brine. The Na^+^, Mg^2+^, Ca^2+^, and Cl^−^ ions have little interference with the boron adsorption. The adsorbents exhibit magnetic separation performance and good cycle life. The results showed that acid-alkali desorption solution has little effect on the adsorbents, and the composite of graphene enhances the adsorption of boron ions. The adsorbents developed in this study are promising to recover boron from low-concentration boron-containing salt-lake brines.

## 1. Introduction

Boron is important for humans, animals, plants, and industry. However, for humans, animals and plants, excessive boron intake can cause various diseases; thus, it is pivotal to control boron levels [1,2,3]. Meanwhile, Boron is a critical raw material for industrial applications, and a large amount of boron is required to develop scientific technologies and manufacture daily necessities [4]. Unfortunately, the techniques for removing and concentrating boron remain a major technical challenge as boron manifests various presences at different concentrations and pH [5,6]. Salt-lake brine contains Li^+^, Na^+^, K^+^, Mg^2+^, Ca^2+^, Cl^−^, SO_4_^2−^, borate, and other anions and cations. Different salt-lake brines have divergent compositions, in which boron content is about 50–1000 mg/L. Although boron is presented in inconsistent forms of anions in solutions under different pH conditions, a large number of experimental results have identified six forms of borate anions, namely (H_3_BO_3_, [B_3_O_3_(OH)_4_]^−^, [B_4_O_5_(OH)_4_]^2−^, [B(OH)_4_]^−^, [B_5_O_6_(OH)_4_]^−^, and [B_6_O_7_(OH)_6_]^2−^) in solutions. In addition, centralized processing in the factory is not feasible due to the cost issue generated by long-distance transportation. To address the needs of boron, several extraction technologies were developed, including acidification, alkali precipitation, extraction, stepwise crystallization, flotation, reverse osmosis, and adsorption [7,8,9,10,11,12]. The adsorption method is promising for boron extraction due to its high selectivity and good separation efficiency. The advantages of simple operation, freedom from harmful substances, recycling, and low costs facilitate efficient fabrications of boron productions [13]. So far, advanced adsorbents with high efficiency and high adsorption capacity have been developed. Most of these adsorbents are functionalized polymers or silicone-based, such as *N*-Methyl-d-glucamine [14], glucose [2], glycidol [15], ortho, or meta-phenol [16]. The mechanisms of extraction involve the complexation between these functional groups and boron [17]. The adsorbent was prepared based on static electricity or ligand exchange. However, the boron adsorbent currently developed has various problems, such as low adsorption capacity, low cycle times, short life, high cost, and an unfriendly environment. It is necessary to further develop low-cost, environmentally friendly, and high-performance adsorbents.

It is important to note that nano-adsorbents have attracted widespread attention due to their excellent adsorption rate and high adsorption capacity. Graphene is a new type of two-dimensional (2D) carbon nanomaterial [18,19,20]. Due to its unique physicochemical properties, the large specific surface area and abundant surface functional groups enable for fabrication of composite adsorbents as matrix material [21]. The Fe_3_O_4_ nanoparticles have good magnetic properties and are combined with graphene to impart magnetic properties for easy separation and purification [22]. Graphene-based magnetic composites are considered excellent adsorbents for removing organic contaminants from aqueous solutions [23]. In addition, it was found that N-adopt graphene can effectively adsorb boron.

In previous studies, we reported the use of polymers as supports for boron adsorbents with excellent adsorption properties. However, the acid and alkali corrosion impedes the promise of solid-liquid separations, where boron can only be adsorbed by functional groups [2,13,24,25]. Herein, aiming to improve stability, we used silicon-based materials that are resistant to acid and alkali corrosion as the main body. Meanwhile, the functional groups and the hydroxyl groups on the surface of graphene are co-adsorbed to strengthen the adsorption process. Magnetic separation effectively solves the solid-liquid separation issues. Four glucose-functionalized inorganic-organic hybrid boron adsorbents were synthesized and further used to remove boron from aqueous solutions. These adsorbents were characterized by IR, XRD, SEM, TEM, XPS, VSM, and BET tests. The effects of pH and salt-lake ions on the adsorption properties of the adsorbents were studied. Based on the results, the adsorption kinetics, isotherm adsorption behavior, and the recycling life of the adsorbents were investigated. The results show that the composite adsorbent can effectively remove and enrich boron. The adsorption-desorption performance of boron adsorbents was evaluated and applied to a real East Taigener salt-lake brine for the validation of scalable processes.

## 2. Results and Discussion

### 2.1. Characterization of the Adsorbents

Figure 1 shows the characterization results of the adsorbents by XRD and FT-IR measurements. The crystalline structure of the Fe_3_O_4_@SiO_2_@mSiO_2_-Glu and Go-Fe_3_O_4_@SiO_2_@mSiO_2_-Glu nanocomposite was determined using an X-ray diffractometer (XRD) (Figure 1a,b). XRD pattern for GO shows a characteristic peak at 23° that is attributed to the graphene phase [21]. The peaks located at 30.5°, 35°, 43.3°, 57°, and 62° are ascribed to Fe_3_O_4_ [23]. Fe_3_O_4_ and SiO_2_ materials and relative intensities of all peaks were confirmed with the standard XRD data of the Joint Committee on Powder Diffraction Standards (JCPDS). The FT-IR spectra of adsorbents, obtained before and after boron adsorption, are shown in Figure 1c,d. The strong absorption peak around about 3350 cm^−1^ for adsorbents represents the vibrations of the (ν_-OH_) groups, which is a general feature of the -OH bond from glucose [26]. The bands at 590 and 1080 cm^−1^ are attributed to the stretching mode of Fe-O [23] and Si-O-Si [27], respectively. In addition, the peaks at 1463, 1240, 1030, and 560 cm^−1^ are ascribed to C=C/C-C stretching mode in graphene sheets [28]. The characteristic stretching vibration bands of -CH_2_ are observed at about 2932 cm^−1^ assigned to propyl chains of the integrated APTEs moieties [29]. The results of the FT-IR analysis indicates that glucose was successfully grafted on Fe_3_O_4_@SiO_2_, while Fe_3_O_4_@SiO_2_-Glu was successfully grafted on graphene.

Figure 2a–d shows the SEM images of the Fe_3_O_4_@SiO_2_-Glu, Fe_3_O_4_@SiO_2_@mSiO_2_-Glu, Go-Fe_3_O_4_@SiO_2_-Glu, and Go-Fe_3_O_4_@SiO_2_@mSiO_2_-Glu. Fe_3_O_4_@SiO_2_-Glu demonstrated uniform-sized spherical particles with a particle size in a range of 400–460 nm, while agglomerations of particles were observed in Fe_3_O_4_@SiO_2_@mSiO_2_-Glu materials. It is suggested that further coating of silica causes agglomeration by increasing adhesion between the particles. Go-Fe_3_O_4_@SiO_2_-Glu display many pellets-like particles on the surface of the wrinkled graphene substrate, which were ascribed to the glucose and SiO_2_ decorated Fe_3_O_4_ nanoparticles. Go-Fe_3_O_4_@SiO_2_@mSiO_2_-Glu shows particle accumulations on the surface of graphene, triggered by the embedding of particles. Compared with Go-Fe3O4@SiO2-Glu, Go-Fe3O4@SiO2@mSiO2-Glu aggregates at the surface of composites due to the weak interactions of materials. The EDS analysis of GO-Fe_3_O_4_@SiO_2_-Glu was listed in the in Appendix A Appendix A. Figure 2e,f shows the TEM image of the Fe_3_O_4_@SiO_2_-Glu and Go-Fe_3_O_4_@SiO_2_@mSiO_2_-Glu. The adsorbents demonstrated a core-shell structure with a graphene sheet layer that has interlayer spacings of 0.262 and 0.275 nm, respectively.

Figure 3a shows the particle size distribution of the adsorbents analyzed by a Malvern laser particle size analyzer. It can be seen that the average particle size and particle size distribution increased sequentially of adsorbents. The results indicate that the second coating of mesoporous SiO_2_ increased the radius of the particles with surging agglomeration between the particles and the sheet. The increase in packing structure and particle size was noticed from the SEM observations. Magnetization curves of the Fe_3_O_4_@SiO_2_-Glu, Fe_3_O_4_@SiO_2_@mSiO_2_-Glu, Go-Fe_3_O_4_@SiO_2_-Glu, and Go-Fe_3_O_4_@SiO_2_@mSiO_2_-Glu microspheres exhibit near-zero coercivity and remanence, suggesting a superparamagnetic behavior as shown in Figure 3b. The saturation magnetization strength is 98.48, 78.72, 3.33, and 21.23 emu/mg for Fe_3_O_4_@SiO_2_-Glu, Fe_3_O_4_@SiO_2_@mSiO_2_-Glu, Go-Fe_3_O_4_@SiO_2_-Glu, and Go-Fe_3_O_4_@SiO_2_@mSiO_2_-Glu materials, respectively. It should be noted that, although the magnetization saturation decreases by adding graphene, the Go-Fe_3_O_4_@SiO_2_-Glu material shows strong magnetization, which indicates the feasibility of magnetic separation. Upon placement of a magnet beside the vial, adsorbents were quickly attracted to the side of the vial within 12 s. Figure 3c,d shows the BET curves of the Fe_3_O_4_@SiO_2_@mSiO_2_-Glu and Go-Fe_3_O_4_@SiO_2_@mSiO_2_-Glu follow the type IV adsorption behavior attributed to the micro-/meso-porosity of the materials. The specific surface area, average pore size, and pore volume of the nanocomposites are summarized in Table 1.

The GO-Fe_3_O_4_@SiO_2_-Glu and Go-Fe_3_O_4_@SiO_2_@mSiO_2_-Glu were analyzed by XPS. The survey spectra (Figure 4a) show the presence of Fe, O, N, C, and Si elements in the GO-Fe_3_O_4_@SiO_2_-Glu and Go-Fe_3_O_4_@SiO_2_@mSiO_2_-Glu composites. In Figure 4b, the spectrum shows two distinguishable peaks at a binding energy of 724, and 709 eV, corresponding to Fe2p3 and Fe2p1 respectively, which is indicative of the presence of magnetite [22]. The O1s XPS spectrum of the GO-Fe_3_O_4_@SiO_2_-Glu and Go-Fe_3_O_4_@SiO_2_@mSiO_2_-Glu at binding energies of 529–536 eV are presented in Figure 4c. The N1 spectrum of the GO-Fe_3_O_4_@SiO_2_-Glu and Go-Fe_3_O_4_@SiO_2_@mSiO_2_-Glu at binding energies of 397–404 eV are shown in Figure 4d. The XPS peaks of C1s centered at the binding energies of 285, 286.3, and 288.4 eV were assigned to the C=C, C-O-H, and C=O, respectively (Figure 4e). The Si2p and Si2p3 XPS spectrum of the GO-Fe_3_O_4_@SiO_2_-Glu and Go-Fe_3_O_4_@SiO_2_@mSiO_2_-Glu at binding energies of 101–105 eV is shown in Figure 4f. It can be seen that the most carbon atoms are sp^2^ hybridized while the C=C peak is predominant, corresponding to the phase of graphene. The evidence of Si-O-Si or O-Fe bonds formed in the composite can be demonstrated by the IR and XPS results.

### 2.2. Adsorption-Desorption Performance of the Adsorbents

The pH of the boron solution governs the ionization of the boron; thereby, presence of different species is determined by the surface charge of the adsorbent [30]. The adsorption behavior of adsorbents was studied in boron solutions with a pH of 4–10. The results in Figure 5a showed that the adsorption capacity of composites was correlated to the pH conditions. With the increase of pH, the adsorption amount increases to a maximum in the alkaline solutions. In the acidic environment, it mainly exists in the form of H_3_BO_3_ (Figure 6a,h) as the pH increases from 4 to 9, resulting in boosting the adsorption capacity. Because the amino and hydroxyl functional groups on the surface of the adsorbents are protonated, the electrostatic interaction with H_3_BO_3_ is weak (Figure 6b). As a result, the adsorption performance is limited. The complexation mechanism between the glucose-functionalized adsorbent and boric acid is shown in Figure 6d. As the pH value of the solution increases, H_3_BO_3_ is continuously converted into B(OH)_4_^−^ and polyanionic, thus yielding a stable tetrahedral complex (Figure 6c) [17]. The adsorption mechanisms between the adsorbents and boron are depicted in Figure 6e–g. When boron is present in the alkaline environment at pH > 9, it is mainly in the form of B(OH)_4_^−^ (Figure 6a,h). The adsorption capacity of the adsorbent is reduced because the polyhydric adsorbent and negatively charged. Electrostatic repulsion between B(OH)_4_^−^ and adsorbents yielded the unsatisfactory behavior of adsorbents [24]. The adsorbents at pH = 9 show adequate adsorption of boron. The results showed that Go-Fe_3_O_4_@SiO_2_@mSiO_2_-Glu materials exhibited improved adsorption capacity compared to other adsorbents. Figure 6i,g presented the XPS analysis of the Fe_3_O_4_@SiO_2_-Glu and Go-Fe_3_O_4_@SiO_2_@mSiO_2_-Glu materials, obtained after boron adsorption. It is observed that the B1s binding energies of the B—O bond of H_3_BO_3_/B(OH)_4_^−^ and B_3_O_3_(OH)_4_/B_3_O_3_(OH)_5_^2−^ in adsorbents were 191.80 eV. Thus, the results verified that boron could be adsorbed by adsorbents. At pH = 9 under concentrations of 5–750 mg/L, the boron adsorption of different adsorbents was shown in Figure 5b. As the concentration of the boron solution increases, the adsorption capacity increases and eventually reaches the adsorption equilibrium. This phenomenon is because the main form of boron at low concentration is H_3_BO_3_/B(OH)_4_^−^ [31]. When the concentration of boron increases, the forms of boron in an aqueous solution were H_3_BO_3_/B(OH)_4_^−^, B_3_O_4_^−^(OH), and B_3_O_3_(OH)_5_^2−^ [25,32], all of which can be adsorbed by adsorbents. As a result, the adsorption amount increases until reaches the adsorption equilibrium. The adsorption capacities of Fe_3_O_4_@SiO_2_-Glu, Fe_3_O_4_@SiO_2_@mSiO_2_-Glu, Go-Fe_3_O_4_@SiO_2_-Glu, and Go-Fe_3_O_4_@SiO_2_@mSiO_2_-Glu were 18.91, 20.99, 16.68, and 23.90 mg/g, respectively. In addition, it can be seen that the adsorption capacity of the Go-Fe_3_O_4_@SiO_2_@mSiO_2_-Glu was the highest among the obtained materials, while the adsorption tests showed an improved kinetic behavior because functional groups enhance the hydrophilic interactions, thus providing an extra diffusion pathway in the macropore domain [21]. The adsorption capacity of obtained adsorbents is much higher than many other silica-supported adsorbents. Table 2 summarizes the adsorption performance comparisons of four adsorbents and some other *N*-methylglucamine and glucose functionalized adsorbents. The adsorption type of boron was analyzed by the Langmuir and Freundlich isotherm adsorption models. The fitting results and constants are listed in Table 3, indicating that the Langmuir (better) and Freundlich models can describe the adsorption process of the adsorbent. The results validate that chemical complexation and physical electrostatic interaction facilitate homogeneous adsorption over the extraction processes [8]. Adsorption is a physicochemical process that describes the transfer of boron to the surface of an adsorbent. To determine the saturation performance, the relationship between adsorption time and adsorption amount was presented in Figure 5c. The adsorption of boron depends on the interfacial properties of the solid-liquid phase, including the contact time, hydrophilic interaction, and diffusion rate of the adsorbent. The adsorption rate increases significantly within 5 min, due to the hydrophilicity of the silica matrix generated by polyhydroxy functional groups. When the adsorption site of the functional group was nucleated by boron, the adsorption rate shows a decrease and finally reaches the adsorption equilibrium. The adsorption equilibrium time of the adsorbent is in a range of 60–90 min attributed to the divergence of composition. To understand the adsorption mechanism, pseudo-first-order, and pseudo-second-order kinetic models were used to fit the experimental data. The parameters are listed in Table 4. The data reveals that the pseudo-second-order kinetic models fitted the experimental values more accurately, which proves that chemisorption dominates the adsorption process. The rapid adsorption kinetics implies that the complexation between the boron and the polyol groups on the adsorbent occurs through chemical adsorption.

Salt-lake brine contains a large number of anions and cations. Among them, the content of Na^+^, Mg^2 +^, Ca^2 +^, and Cl^−^ ions in the salt lake containing the boron is very high. Therefore, the effects of interfering ions Na^+^, Mg^2+^, Ca^2+^, and Cl^−^ on boron adsorption were investigated. The boron adsorption performance of the adsorbent was evaluated at the concentration of interfering ions five times higher than boron ions (500 mg/L). The adsorption data are compared with the blank as shown in Figure 5d. The results show that the adsorbents have excellent selectivity of boron with evidence that the ion interference has little effect. The proof-of-concept investigation demonstrated that the adsorbents developed in this work can selectively adsorb boron in a multi-ion salt-lake brine system.

To study the cycle stability of the adsorbents, the boron on the adsorbent was eluted with HCl (0.5 mol/L) solution, then washed with water, and neutralized with NaOH (0.1 mol/L) (the process is repeated about three times and the solid adsorbent is separated by external magnetic field). After drying, the materials were subjected to secondary adsorption experiments. Over the adsorption-desorption processes, the boron adsorption capacity of obtained adsorbents was slightly reduced (Figure 5e). The results show that the materials have a good service life and recycling performance.

Figure 5f shows the static adsorption of the Go-Fe_3_O_4_@SiO_2_@mSiO_2_-Glu for the East Taigener salt-lake brine. The composition of East Taigener salt-lake brine is given in Appendix A. For such brine with a boron concentration of 807 mg/L, the Go-Fe_3_O_4_@SiO_2_@mSiO_2_-Glu material delivered a boron adsorption capacity of 24.84 mg/g. The adsorption process followed second-order kinetics models, as the calculations revealed (Table 5).

## 3. Materials and Methods

### 3.1. Materials, Physicochemical Measurements, and Method of Determination of Boron

Graphite (Go), glucose (Glu), KMnO_4_, NaNO_3_, Cetyltrimethyl Ammonium Bromide (CTAB), H_2_O_2_, H_2_SO_4_ (98%), FeCl_3_·6H_2_O, NaOH, HCl, Ammonia water, 3-aminopropyltriethoxysilane (APTEs), ethanol, sodium acetate, ethylene glycol, tetraethyl orthosilicate (TEOS), Boric acid, NaCl, MgCl_2_, CaCl_2_ were purchased from Aladdin Ltd. (Shanghai, China) and Tianjin Zhiyuan Chemical Reagent Co., Ltd. (Tianjin, China).

All of the organic solvents used in this study were dried over appropriate drying agents and distilled prior to use. The aqueous solutions were prepared with Milli-Q water for the adsorption-desorption experiments.

Boron analyses were performed by UV-vis spectroscopy using the azomethine-H method [38] Azomethine-H(4-Hydroxy-5-((2-hydroxybenzylidene)amino)naphthale-ne-2,7-disulfonic acid), acetic acid, EDTA-2Na, ascorbic acid, and ammonium acetate were main reagents for the analysis of boron.

IR spectrum was taken with a Shimadzu IR Prestige-21 FT-IR spectrophotometer. The surface morphology of the adsorbent was visualized by an SEM (JEOL, JSM-5600V, Tokyo, Japan) and a SU-8010 TEM instrument (Hitachi, Tokyo, Japan). X-ray diffraction analysis (XRD) measurement was performed on a Bruker D8 advance (40 kV, 40 mA) with Cu-Kα radiation (λ = 1.5406 nm), and the diffraction patterns were collected in the 2*θ* ranging from 10° to 80° at a scanning rate of 1.2°/s. X-ray Photoelectron Spectroscopy (XPS) was conducted on a thermoelectric instrument (ESCALAB 250Xi, Thermo Scientific, Waltham, MA, USA) with Al K Alpha 1486.6 eV. Magnetization measurements were performed on a superconducting quantum interference device (SQUID) magnetometer at 300 K (VSM). The concentration of boron in the solution was gauged by applying an ultraviolet and visible spectrophotometer (T6, Beijing Purkinje General Instrument Co., Ltd., Beijing, China).

### 3.2. Synthesis of the Fe_3_O_4_@SiO_2_-Glu, Fe_3_O_4_@SiO_2_@mSiO_2_-Glu, Go-Fe_3_O_4_@SiO_2_-Glu, and Go-Fe_3_O_4_@SiO_2_@mSiO_2_-Glu

The preparation of Fe_3_O_4_@SiO_2_-Glu, Fe_3_O_4_@SiO_2_@mSiO_2_-Glu, Go-Fe_3_O_4_@SiO_2_-Glu, and Go-Fe_3_O_4_@SiO_2_@mSiO_2_-Glu was illustrated in Figure 1. It is worth noting that the addition of an ammonia solution may make N doped with graphene and improve the adsorption capacity of boron.

#### 3.2.1. Preparation of Fe_3_O_4_

Solvothermal synthesis of Fe_3_O_4_ nanoparticles: 2.72 g of FeCl_3_·6H_2_O was dissolved in 80 mL of ethylene glycol, 7.2 g of NaAc was added, continuously stirred and sonicated, transferred to the reaction vessel and reacted at 180 °C for 12 h. Magnetically separated and continuously washed with water and dried at 60 °C.

#### 3.2.2. Preparation of Fe_3_O_4_@SiO_2_

A quantity of 0.1 g of Fe_3_O_4_ were homogeneously dispersed in a mixture of 40 mL of ethanol, 10 mL of deionized water and 1.2 mL of 28 wt% NH_3_·H_2_O, followed by the addition of 0.4 mL of TEOS. After vigorous stirring at room temperature for 6 h, the obtained Fe_3_O_4_@SiO_2_ microspheres were separated with a magnet and washed repeatedly with ethanol and water, and dried at 60 °C.

#### 3.2.3. Preparation of Fe_3_O_4_@SiO_2_@mSiO_2_

A quantity of 0.1 g of Fe_3_O_4_@SiO_2_ particles were uniformly dispersed in 60 mL of ethanol by ultrasonic treatment, and then 1.2 mL of 28 wt% NH_3_·H_2_O was added to form a solution A. Then, 0.3 g of CTAB was added to 80 mL of H_2_O to form solution B. After, Solution B was mixed with Solution A for 6 h with vigorous stirring. Then, 0.43 mL of TEOS was added dropwise to the solution. After mechanical stirring for 6 h, the obtained granules were separated with a magnet and washed with deionized water. Finally, the purified sample was redispersed in 80 mL of acetone and refluxed at 80 °C for 24 h to remove the CTAB template. The extraction was repeated three times. The resulting powder was then washed with water and ethanol and dried at 60 °C

#### 3.2.4. Preparation of Fe_3_O_4_@SiO_2_-NH_2_ and Fe_3_O_4_@SiO_2_@mSiO_2_-NH_2_

Fe_3_O_4_@SiO_2_-NH_2_ was synthesized in the same way as Fe_3_O_4_@SiO_2_@mSiO_2_-NH_2_. 0.1 g Fe_3_O_4_@SiO_2_ or Fe_3_O_4_@SiO_2_@mSiO_2_ was dispersed in 20 mL toluene by ultrasound, then 1.5 mL 3-aminopropyl trimethoxysiloxane was added, and then reflux stirred for 20 h at 110 °C. The obtained powder was washed with water and ethanol and dried at 60 °C.

#### 3.2.5. Preparation of Fe_3_O_4_@SiO_2_-Glu and Fe_3_O_4_@SiO_2_@ mSiO_2_-Glu

Fe_3_O_4_@ SiO_2_-Glu and Fe_3_O_4_@SiO_2_@ mSiO_2_-Glu were synthesized in the same method. 0.1 g of Fe_3_O_4_@SiO_2_-NH_2_ or Fe_3_O_4_@SiO_2_@mSiO_2_-NH_2_ were ultrasonically dispersed in 20 mL methanol, then 1.5 g of glucose were added, respectively. Then stirred at 45 °C for 12 h. The obtained powder (Fe_3_O_4_@SiO_2_-Glu or Fe_3_O_4_@SiO_2_@ mSiO_2_-Glu) was washed with ethanol and water and dried at 45 °C.

#### 3.2.6. Preparation of Graphene

Preparation Graphene oxide (Go) was prepared by the modified hummers method according to the literature [20].

#### 3.2.7. Preparation of Fe_3_O_4_-Graphene

Fe_3_O_4_-graphene composites were prepared by the one-pot solvothermal synthesis method. Generally, Go (0.30 g) was ultrasonically dispersed to ethylene glycol (30 mL) for two hours. Then, sodium acetate (1.32 g) and FeCl_3_·6H_2_O (0.5 g) were added to the mixture solution and stirred for twenty minutes. Finally, the mixture solution was transferred to the reaction kettle and heated at 180 °C for 8 h. The final product was washed with water and ethanol three times, respectively, and then vacuum dried at 60 °C for 12 h.

#### 3.2.8. Preparation of Go-Fe_3_O_4_@SiO_2_ and Go-Fe_3_O_4_@SiO_2_@mSiO_2_

The Go-Fe_3_O_4_@SiO_2_ and Go-Fe_3_O_4_@SiO_2_@mSiO_2_ synthetic steps refer to Section 3.2.2 and Section 3.2.3. Performed as described in reference [23]. 

In total, 0.1 g of Go-Fe_3_O_4_ were homogeneously dispersed in a mixture of 40 mL of ethanol, 10 mL of deionized water and 1.2 mL of 28 wt% NH_3_·H_2_O, followed by the addition of 0.3 mL of TEOS. After vigorous stirring at room temperature for 6 h, the obtained Go-Fe_3_O_4_@SiO_2_ microspheres were separated with a magnet and washed repeatedly with ethanol and water, and dried at 60 °C.

A quantity of 0.1 g of Go-Fe_3_O_4_@SiO_2_ particles were uniformly dispersed in 60 mL of ethanol by ultrasonic treatment, and then 1.2 mL of 28 wt% NH_3_·H_2_O was added to form solution A. Then, 0.3 g of CTAB was added to 80 mL of H_2_O to form solution B. After, solution B was mixed with Solution A for 6 h with vigorous stirring. Then, 0.3 mL of TEOS was added dropwise to the solution. After mechanical stirring for 6 h, the obtained particles were separated with a magnet and washed with deionized water. Finally, the purified sample was redispersed in 80 mL of acetone and refluxed at 80 °C for 24 h to remove the CTAB template. The extraction was repeated three times. The resulting powder was then washed with water and ethanol and dried at 60 °C.

#### 3.2.9. Preparation of Go-Fe_3_O_4_@SiO_2_-NH_2_ and Go-Fe_3_O_4_@SiO_2_@mSiO_2_-NH_2_

The synthesized magnetic nanoparticles of Go-Fe_3_O_4_@SiO_2_ (2 g) or Go-Fe_3_O_4_@SiO_2_@mSiO_2_ (2 g) were added to 10 mL of toluene and sonicated for 40 min. Then, 3 mL of (3-aminopropyl) triethoxysilane was admixed to the above mixture while stirring and the resulting mixture was refluxed for 40 h. The final product was washed with methanol. Go-Fe_3_O_4_@SiO_2_-NH_2_ and Go-Fe_3_O_4_@SiO_2_@mSiO_2_-NH_2_ were obtained, respectively.

#### 3.2.10. Preparation of Go-Fe_3_O_4_@SiO_2_-Glu and Go-Fe_3_O_4_@SiO_2_@mSiO_2_-Glu

The synthesis steps of Go-Fe_3_O_4_@SiO_2_-Glu and Go-Fe_3_O_4_@SiO_2_@mSiO_2_-Glu were similar to 3.25. A quantity of 0.1 g of Go-Fe_3_O_4_@SiO_2_-NH_2_ or Go-Fe_3_O_4_@SiO_2_@mSiO_2_-NH_2_ were dispersed in 20 mL of methanol ultrasound for 30 min, then 1.5 g of glucose were added, respectively. And stirred at 45 °C for 12 h. The powder (Go-Fe_3_O_4_@SiO_2_-Glu and Go-Fe_3_O_4_@SiO_2_@mSiO_2_-Glu) was washed repeatedly with ethanol and water and dried at 45 °C.

### 3.3. Boron Adsorption-Desorption Experiments

To evaluate adsorption performance, the kinetic behavior of obtained adsorbents was analyzed based on aqueous boron solutions. The effects of pH (4–10), initial concentration of boron (5–750 mg/L), adsorption time (5–720 min), and ionic (Na^+^, Mg^2+^, Ca^2+^, and Cl^−^) on the adsorption properties of the adsorbents were discussed. In this study, 0.2 g of adsorbent was added to 50 mL of known concentration of boron, in the constant temperature oscillator and the shaking speed was 200 rpm at 25 °C. The dispersed adsorbents can be easily separated by an external magnet. For the desorption experiments, the adsorbents were added to 100 mL 0.1 M HCl, after the mixture was shaken for 1 h, the solid adsorbents were separated by an external magnet and added to 20 mL 0.5 M NaOH after the mixture was shaken for 0.2 h, the solid adsorbents were separated by an external magnet. This process is repeated about three times, washed with deionized water, dried, and the secondary adsorption experiment is carried out. The adsorption capacity (mg/g) was calculated from the *Q* = (*C*_0_ − *C*_e_)V/Mm, where *C*_0_ and *C*_e_ (mg/L) are the initial and equilibrium concentration of boron in the solution, respectively. *V* is the volume of the solution (L), and m is the mass of adsorbents. M is the mole weight of H_3_BO_3_ (g/mol). Adsorption isotherms and kinetics were fitted to the corresponding equations [39] and listed in Appendix A.

## 4. Conclusions

Four adsorbents Fe_3_O_4_@SiO_2_-Glu, Fe_3_O_4_@SiO_2_@mSiO_2_-Glu, Go-Fe_3_O_4_@SiO_2_-Glu, and Go-Fe_3_O_4_@SiO_2_@mSiO_2_-Glu were prepared by glucose-functionalized silica-coated Fe_3_O_4_ and glucose-functionalized silica onto the Fe_3_O_4_ embedded with graphene. The performance of boron adsorption was investigated under different conditions. The results showed that obtained adsorbents demonstrated high adsorption capacities. Graphene and polyhydroxy functional groups enhanced the adsorption properties of boron. The adsorption capacity of Go-Fe_3_O_4_@SiO_2_@mSiO_2_-Glu was up to 23.9 mg/g, which is higher than current silica-supported adsorbents. The presence of Na^+^, Mg^2+^, Ca^2+^, and Cl^−^ in boron solution has little effect on boron adsorption, which was attributed to the high selectivity of adsorbents. After 5 adsorption-desorption cycles, the adsorbents showed stable performance. The adsorption kinetics were analyzed based on the Langmuir model and the Pseudo-second-order kinetic model. Moreover, the adsorbents can be extracted from the solution by external magnets. The results demonstrate that the composite adsorbent is promising for boron adsorptions (24.84 mg/g) from salt-lake brine.

## Data Availability

Not applicable.

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
