# Peer review of "Magnetic Separation of Oxoacid of Boron from Salt-Lake Brine by Synergistically Enhanced Boron Adsorbents of Glucose-Functionalized SiO2 and Graphene"

_ijms, 2022, doi:10.3390/ijms231911356_

Round 1

Reviewer 1 Report

Article "Magnetic separation of boron from salt lake brine by synergistically enhanced boron adsorbents of glucose-functionalized SiO2 and graphene" Authors: Qinglong Luo , Xueying Wang , Mingzhe Dong , Xueli Huang, Zhijian Wu , Jun Li are dedicated to topical topics of the modern world: rational use of natural resources. The article corresponds to the subject of the journal "IJMS". The article is well structured, written in a clear and understandable language, the conclusions are logical, the literature corresponds to the stated topic. However, there are some notes:

1.

The authors use the phrase "separation of boron from salt" in the title of the article. The first impression is that the work is devoted to the sorption of boron (elemental). While boron exists in salt brine as any oxoacid of boron. I propose to change the title of the article.

Replace the phrase "separation of boron from salt " with "separation of oxoacid of boron from salt ".

Further in the text of the article, the sorption of boron is often mentioned, and studies are devoted to the sorption of oxoacid of boron. A discussion of the form in which boron is in brine begins only in section "2.2 Adsorption–desorption performance of the adsorbents".

I advise the authors to move the discussion of the form of boron in the composition of the brine from this section to the beginning of the article, for example, to the introduction. At the same time, in the introduction it is necessary to mention (discuss) the chemical composition of the salt lake, indicate the proportion of oxoacid of boron.

2.  " Figure 4. Legend - text labels is blurry. Please fix it.

Reviewer 2 Report

The manuscript titled “Magnetic separation of boron from salt lake brine by synergistically enhanced boron adsorbents of glucose-functionalized SiO2 and graphene” by Luo et al. reported the design of magnetic adsorbents for the recovery of boron from salt-lake brine. In particular, four adsorbents with the compositions, Fe3O4@SiO2-Glu, Fe3O4@SiO2@mSiO2-Glu, Go-Fe3O4@SiO2-Glu, and Go-Fe3O4@SiO2@mSiO2-Glu were synthesized and evaluated for the adsorption of boron. This type of study is important for the effective recovery of valuable resources from seawater. However, I have some comments which should be addressed before the acceptance of the paper for publication.

1.     Part of the significance of the study is the magnetic properties of the adsorbents which simplifies the separation and recovery of the materials for further reuse. However, I did not find any section in the manuscript where this was implemented to study the material reusability. Rather the authors used HCl (0.5 mol/L) and NaOH (0.5 mol/L) for the recovery. I suggest to include the magnetic separation and reusability.

2.     The Figures and scheme should appear in the order which they were discussed in the text. For instance, scheme 1 appeared on page 3, while the discussion appeared much later on page 12. In addition, the quality of the Figures should be improved. The texts in Figure 1 are not readable. This applies to Figure 4 and the inset in Figure 6.

3.     The use of acronyms such as Fe3O4@SiO2-Glu and so on all through the text without defining each one in the sentence they first appear is somewhat confusing. Authors should define each acronym where it first appears or provide an appendix with all the acronyms and their meaning at the end of the manuscript.

4.     On page 4, the authors wrote “Go-Fe3O4@SiO2-Glu display many pellets-like particles on the surface ............., which were ascribed to the glucose and SiO2 decorated Fe3O4 nanoparticles”. This statement should be supported by checking the elemental distribution/mapping on the surface using techniques such as EDS.

5.     The manuscript needs a thorough English revision to fix some grammatical and punctuation errors. I have highlighted a few here:

I.                On page 4, “…….. The adsorbents are core-shell structure and its composite structure with sheet graphene…….”. This should be corrected to graphene sheet.

II.              On page 6, “Figure 3a shows the particle………… size analyzer,”. This should end with a full stop and not comma.

III.            On page 6, “It with the scanning electron microscope of the adsorbents corresponds to each other.” This sentence is not clear and should be rewritten.

IV.            On page 6, “Figure c,d shows that the adsorption……..” The Figure number should be indicated such as Figure 3c,3d if referring to Figure 3.

V.              On page 7, “Therefore, the adsorption of adsorbent in boron solution……….” This sentence should be revised.

VI.            On page 8, “There is electrostatic repulsion between B(OH)4- [24]”. Between B(OH)4- and what? The sentence is incomplete.

VII.         On page 8, “This indicates that the adsorbent is adsorbed by chemistry and physics the process………”. This sentence is wrong as the adsorbent is not adsorbed, rather it is doing the adsorption. Also, the use of chemistry and physics makes no meaning and should be removed.

VIII.       On page 8, “while the rapid adsorption kinetics indicate that the adsorption process is mainly through chemistry adsorption”. Is it chemistry adsorption or chemisorption?

IX.            On page 9, “a large amount of anion and cation.” It should be anions and cations (plural).

X.              On page 9, “Based on keeping the concentration of boron …………. respectively.” This sentence is incomplete.

XI.            On page 9, “and it can adsorption boron in salt lake brine.” The sentence is grammatically wrong and should be corrected.

Round 2

Reviewer 2 Report

The authors have addressed all the comments raised by the reviewer. Hence the paper can be accepted in its current form.